🔓 | **Open Peer Review** | Genetics and Molecular Biology | Research Article

# Isolation of a $bla_{NDM-1}$-positive strain in Israel predating the earliest observations from India

Frédéric Grenier,[1] Vincent Baby,[1] Sarah Allard,[2] Simon Lévesque,[2,3] François Papale,[2] Richard Sullivan,[4] Hannah L. Landecker,[5] Paul G. Higgins,[6,7] Sébastien Rodrigue,[1] Louis-Patrick Haraoui[2,8]

**ABSTRACT** $bla_{NDM}$, the most prevalent carbapenemase among carbapenem-resistant Enterobacteriaceae, is thought to have emerged in India, as its initial detection in 2008 was linked to this country, and subsequent retrospective surveys had so far established the earliest $bla_{NDM}$-positive strains to be isolated in India in 2005. Molecular dating and analyses suggest $bla_{NDM}$ emerged within *Acinetobacter* species decades prior to 2005 on a Tn*125* transposon. Despite early reports of elevated rates of carbapenem-resistant *Acinetobacter* species in Israel starting in the 1990s, limited molecular data are available from this location. We searched for $bla_{NDM}$ among *Acinetobacter* species isolated in Israel between 2001 and 2006. One *A. junii* strain, Ajun-H1-3, isolated in January 2004, carried $bla_{NDM-1}$ within a Tn*125*-like transposon on a 49-kb plasmid, pNDM-Ajun-H1-3, making Ajun-H1-3 the earliest NDM-positive isolate observed to date. The pNDM-Ajun-H1-3 plasmid matched numerous BJ01-like NDM-positive plasmids identified from 2005 onward in *Acinetobacter* species as well as Enterobacterales. These results indicate the need for further retrospective work on global strain archives to shed light on the conditions favoring the emergence as well as subsequent evolution and spread of $bla_{NDM}$.

**IMPORTANCE** This study presents the earliest observation of $bla_{NDM-1}$, isolated in a geographical region distant from where it is believed to have originated. In doing so, this study provides novel insights into the emergence and spread of $bla_{NDM}$, the most prevalent carbapenemase among carbapenem-resistant Enterobacteriaceae, and its associated mobile genetic elements. It also sheds light on the conditions that foster the evolution of antimicrobial resistance, one of the greatest public health challenges we face.

**KEYWORDS** Antimicrobial resistance, *Acinetobacter junii*, BJ01-like plasmid, Tn*125* transposon, $bla_{NDM}$

Address correspondence to Louis-Patrick Haraoui, Louis.Patrick.Haraoui@USherbrooke.ca.

The authors declare no conflict of interest.

See the funding table on p. 8.

Antimicrobial resistance (AMR) is a major global public health issue, with AMR deaths surpassing those of HIV and malaria in terms of infectious disease-related mortality (1). In 2018, the World Health Organization (WHO) issued its priority list for the discovery, research, and development of new treatments for antibiotic-resistant bacteria. All priority 1 critical pathogens—*Acinetobacter baumannii*, *Pseudomonas aeruginosa,* and Enterobacteriaceae—share a common trait: resistance to carbapenems, an antibiotic class within the broad beta-lactam family (2).

Carbapenemases constitute the most common enzymes through which bacteria become resistant to carbapenems. One such enzyme family, New Delhi metallo-beta-lactamases (NDM) encoded by variants of the $bla_{NDM}$ gene, have become the most prevalent enzymes among carbapenemase-producing Enterobacteriaceae clinical isolates worldwide, in spite of being reported more recently than the other predominant

carbapenemases (3). Numerous lines of evidence point to $bla_{NDM}$ originating in an "*Acinetobacter* background" (4, 5). Despite this association, the vast majority of *Acinetobacter baumannii* become carbapenem-resistant through acquisition of Class D beta-lactamases such as $bla_{OXA-23}$, and only in a minority of cases do carbapenem-resistant *A. baumannii* carry $bla_{NDM}$ (6).

One of the global regions initially affected by the rise and spread of carbapenem resistance in *Acinetobacter* species (spp.) was the Middle East. Reports of rapidly increasing rates of carbapenem-resistant *Acinetobacter* spp. date back to the 1990s in Israel (7–9). The earliest NDM-positive *Acinetobacter* strain observed from the Middle East was, to this day, an *A. pittii* isolated in 2006, 1 year after the oldest known isolates from India (5), in a Turkish patient with no travel history outside Turkey (10).

Given the documented early and elevated rates of carbapenem-resistant *Acinetobacter* spp. in Israel, the apparent role of *Acinetobacter* spp. in the emergence of $bla_{NDM}$, and the lack of retrospective analyses among isolates from the early 2000s, we investigated the presence of $bla_{NDM}$ among 198 *Acinetobacter* spp. clinical isolates from Israel between 2001 and 2006. Among them, we have identified the earliest NDM-positive isolate observed to date.

## MATERIALS AND METHODS

### Strains

We obtained 198 clinical *Acinetobacter* spp. isolated in Israel between 2001 and 2006 from three archives: 1) directly from Chaim Sheba Medical Center, Tel Hashomer, Israel (*n* = 140); 2) from JMI Laboratories, who, as part of their SENTRY Antimicrobial Surveillance Program, still held 37 isolates also from Chaim Sheba Medical Center, distinct from those we obtained directly from the medical center; and 3) 21 isolates from International Health Management Associates Inc., collected from two different anonymized hospitals. We also obtained, from the Barcelona Institute for Global Health Foundation, two NDM-positive isolates for comparative purposes: an *A. pittii* (JVAP02) and an *A. lactucae* (JVAP01), both isolated in Turkey in 2006 and 2009, respectively (10, 11).

### Whole-genome sequencing

All isolates were grown in lysogeny broth (LB) at 37°C overnight. DNA libraries were prepared from extracted gDNA using the NEBNext Ultra II FS DNA Library Prep Kit for Illumina (NEB). DNA was purified and size-selected using AMPure XP beads (Beckman Coulter) and quantified using Quant-iT PicoGreen dsDNA assay (Thermo Fisher). The quality and size distribution of the DNA were assessed on a Fragment Analyzer using the HS NGS Fragment Kit (Agilent). The pooled samples were then sequenced on a NovaSeq 6000 (Illumina) with 250-bp paired-end read sequencing.

A subset of strains, including Ajun-H1-3, was also sequenced using an Oxford Nanopore Technologies (ONT) R10.4 Flow Cell on a MinION Mk1B. Extracted gDNA was treated with the NEBNext Ultra II End Repair/dA-Tailing Module (NEB). Barcodes from the Native Barcoding Expansion 1–12 & 13–24 from ONT were ligated using the NEBNext Ultra II Ligation Module (NEB). DNA was purified using AMPure XP beads (Beckman Coulter). The DNA from different barcoded samples was pooled, and the adapter AMII (ONT) was ligated using the NEBNext Ultra II Ligation Module (NEB).

### Data analysis and bioinformatics

For Illumina reads, the quality assessment and trimming were done using fastp 0.21.0 with --cut_right --cut_window_size 4 --cut_mean_quality 20 --length_required 30 --detect_adapter_for_pe (12). Assemblies were made using Unicycler 0.4.9 (13) with the trimmed Illumina short reads and ONT long reads when available. Contigs were filtered to retain only those above 500 bp.

Taxonomic identification was made on the assemblies using Kraken 2 (2.0.9-beta) (14). Antibiotic resistance genes were found using ResFinder 4.1 (15). Detected $bla_{OXA}$ variants were curated using BLDB (16). Assemblies were annotated with Prokka 1.14.5 (17) using the additional databases Pfam, TIGRFAM, and the $bla_{OXA}$ variants present in BLDB (16). Plasmid alignments were made using AliTV (18).

## Conjugation experiments

### Strains and growth conditions

We used Ajun-H1-3 as the donor strain. The recipient strains were spontaneous resistant mutants to rifampicin, which were generated using *Escherichia coli* MG1655 (19). The strains were grown on LB and on LB agar medium with the appropriate antibiotics at the following working concentrations: rifampicin 66.7 µg/mL; meropenem 9 µg/mL.

### Conjugation assay

The donor and recipient strains were subcultured from frozen stocks for 18 hours at 37°C prior to conjugation experiments. For both strains, the equivalent of 100 µL of the culture at an OD600 of 1.0 was spun and resuspended in 50 µL of fresh LB to remove antibiotics. For every conjugation experiment, 2.5 µL of both washed strains was mixed, and the conjugation mixture droplet was put on a 500 µL bottom of LB agar in a 2-mL screw cap tube. Negative controls were also prepared by adding only one of the strains. The tubes were left uncapped for 1 hour under the biological hood for the mixture to dry. The tubes were then capped and placed at room temperature or at 37°C for 2 hours or 16 hours for the conjugation to occur. Subsequently, 500 µL of LB was added to the tubes, and they were vortexed for 10 seconds. One hundred microliters of the resuspended conjugation mixture was then used to prepare tenfold dilutions with 1X PBS. Five microliters of the dilutions was spotted in triplicates on LB agar plates with the appropriate antibiotics. The plates with the dilution droplets were left to dry under the hood for a few minutes before being incubated at 37°C for 16 hours, after which the colony-forming units (CFUs) were counted.

## RESULTS

Strain Ajun-H1-3 is a clinical *Acinetobacter junii* isolate collected in January 2004 from a blood culture at Chaim Sheba Medical Center in Tel Hashomer, Israel. Ajun-H1-3 possesses two carbapenemases, $bla_{OXA-58}$ and $bla_{NDM-1}$, the latter on plasmid pNDM-Ajun-H1-3, which was distinct from the $bla_{OXA-58}$-encoding plasmid. Ajun-H1-3 is the earliest known $bla_{NDM}$ isolate to date, and it was the sole NDM-positive isolate of the data set. No records were accessible to obtain further details about the patient from whom it was recovered, including potential travel history outside Israel.

In pNDM-Ajun-H1-3, $bla_{NDM-1}$ was embedded on a Tn*125*-like transposon containing three full copies of IS*Aba125* (Fig. 1A), instead of the commonly reported two copies among other early NDM-positive isolates (20). An *aph(3')-VIa* aminoglycoside resistance gene was the sole gene found in between the two IS*Aba125* downstream of $bla_{NDM-1}$ (Fig. 1A). This transposon configuration including $bla_{NDM-1}$, three full copies of IS*Aba125,* and the *aph(3')-VIa* resistance gene has only been reported in two other instances in the nr/nt database: on a 265-kb plasmid in an *A. baumannii* isolated from the trachea of a duck in Guangdong province, China, in December 2017 (GenBank accession number CP048828.1); and on a 123-kb plasmid in an *A. variabilis* isolated from a human wound/abscess in the United States in May 2021 (GenBank accession number CP104653.1). pNDM-Ajun-H1-3, which is a 49-kb plasmid (exact size: 49,163 base pairs), was otherwise highly dissimilar to these two plasmids.

Indeed, pNDM-Ajun-H1-3 was found to match numerous other NDM-positive plasmids termed pNDM-BJ01-like, based on their similarity to a plasmid initially reported in an *A. lwoffii* strain from November 2010 in China (21). Since this initial description, pNDM-BJ01-like plasmids have been shown to be widely circulating among

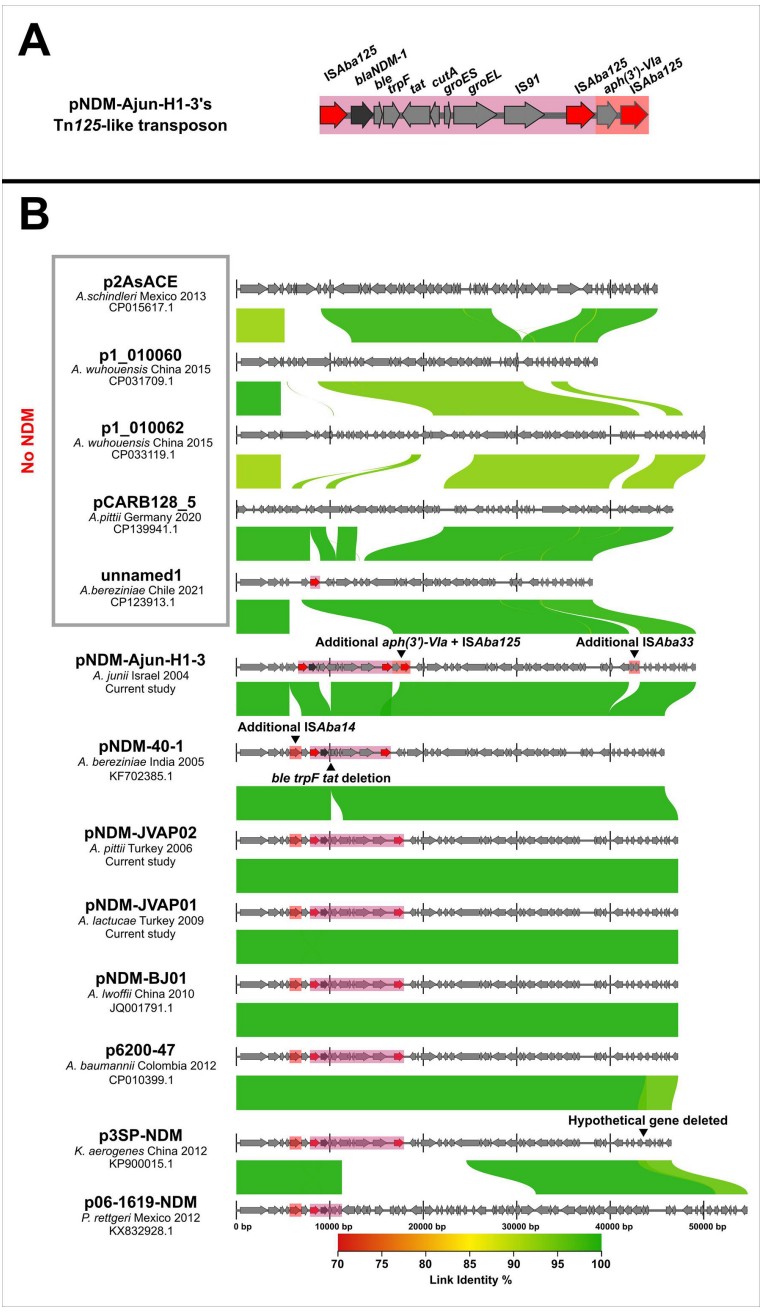

**FIG 1** Comparison and alignment of certain BJ01-like plasmids in *Acinetobacter* spp. and Enterobacterales. Panel 1 (A) Genetic structure of the Tn*125*-like transposon carrying *bla*NDM-1 within pNDM-Ajun-H1-3, containing an additional *aph(3')-VIa* gene and additional IS*Aba125* insertion sequence compared to other Tn*125* transposons. Panel 1 (B) The top five plasmids are the only BJ01-like plasmids deposited in GenBank lacking *bla*NDM (presented in chronological order from top to bottom). Below these, some of the *bla*NDM-containing BJ01-like plasmids are also present in chronological order from top to bottom, starting with the earliest known observation of *bla*NDM-1 reported as part of this study. A color gradient of the link identity percentage presents genetic sequence similarities. Alignments demonstrate dissemination of highly conserved plasmids: pNDM-Ajun-H1-3 is 99.99% similar over 97% of pNDM-40–1's sequence and 99.98% similar over 97% of pNDM-BJ01's sequence. The species identification, country, year of isolation, and GenBank accession number are listed underneath the plasmid name. Although data for pNDM-JVAP01 were already available (11), the assembly presented here is derived from our own sequencing.

*Acinetobacter* spp., particularly among non-*baumannii Acinetobacter* (NbA), including pNDM-40–1 in an *A. bereziniae* from India from 2005 (4); pNDM-JVAP02 and pNDM-JVAP01, included in this report (from an *A. pittii* and an *A. lactucae* isolated in Turkey in 2006 and 2009, respectively) (10, 11); numerous *Acinetobacter* spp. from China from 2009 onward (22), as well as elsewhere around the world. pNDM-BJ01-like plasmids have also been identified in Enterobacterales clinical isolates in China and in Mexico in one *Klebsiella aerogenes* and one *Providencia rettgeri,* respectively (23, 24).

A BLAST search using the nr/nt database combined with a literature review (2 April 2024) revealed a total of 51 BJ01-like plasmids (23). Of these, 46 carried $bla_{NDM}$ among 36 NbA, eight *A. baumannii*, one *Klebsiella aerogenes,* and one *Providencia rettgeri*. These included 42 $bla_{NDM-1}$ among 32 NbA, eight *A. baumannii*, one *Klebsiella aerogenes,* and one *Providencia rettgeri*; two strains isolated in South Korea in 2015 (CP137748.1: one *Acinetobacter sp.* 16, recently renamed *A. higginsii* (25); and OR508935.1: one *A. junii*) reported as carrying $bla_{NDM-1}$, but whose $bla_{NDM}$ sequence does not correspond to any known $bla_{NDM}$ variant, likely constituting a new $bla_{NDM}$ variant as they each contain an identical single amino acid difference to $bla_{NDM-1}$ (Arg81Lys); one *A. nosocomialis* with $bla_{NDM-1}$ containing a frameshift mutation (one base-pair deletion; MK053934.1); one *A. lwoffii* whose BJ01-like plasmid (KM210086.1) harbored $bla_{NDM-14}$, a single amino acid change compared to $bla_{NDM-1}$ (Asp130Gly) (26). Five BJ01-like plasmids lacking $bla_{NDM}$ were all isolated in NbA (one in Mexico in 2013; two in China in 2015; one in Germany in 2020; one in Chile in 2021), several years after the earliest observations of $bla_{NDM}$-containing BJ01-like plasmids. To our knowledge, pNDM-BJ01-like plasmids are the only NDM-positive plasmids common to both *Acinetobacter* spp. and Enterobacterales. BJ01-like plasmid alignments, including pNDM-Ajun-H1-3, pNDM-JVAP01, and pNDM-JVAP02, as well as the five lacking $bla_{NDM}$ are presented in Fig. 1B.

Results from conjugation experiments using Ajun-H1-3 as the donor strain confirmed its ability to transfer pNDM-Ajun-H1-3 to an *E. coli* recipient. Exconjugants were only detected with a conjugation incubation lasting 16 hours (absent after 2 hours of incubation). The corresponding transfer rates were 3.9 E-8 and 2.8 E-6 exconjugants/recipient at room temperature and 37°C, respectively.

## DISCUSSION

Ajun-H1-3, an *A. junii* isolated from a blood culture in January 2004, is the earliest NDM-positive isolate observed, predating the seven NDM-positive *Acinetobacter* spp. (six *A. baumannii* and one *A. bereziniae*) from Tamil Nadu, India, from 2005, which were considered the earliest isolates until now (5).

$bla_{NDM-1}$ was initially detected in 2008 in a patient who was receiving care in Sweden following a trip to India, during which he had been hospitalized in Ludhiana (Punjab) and in New Delhi (27). Since then, $bla_{NDM}$ has become the most prevalent carbapenemase among carbapenem-resistant Enterobacteriaceae, also finding its way into numerous other γ-Proteobacteria such as *Pseudomonas* spp (3). Over 60 enzyme variants exist, with new ones still being described (http://www.bldb.eu/, last accessed 24 April 2024). $bla_{NDM}$ has been implicated in human infections (28), identified in domestic and farm animals (29, 30), and spread to environments devoid of human presence, such as High Arctic soil ecosystems (31). Retrospective analyses have consistently shown the earliest NDM-positive strains in any given geographical region to be *Acinetobacter* spp.

The significance of the NDM-positive-Ajun-H1-3 identified in this study lies in its implications for our understanding of the origins of $bla_{NDM}$: it highlights its potentially environmental and NbA origins, as well as the importance of mobile genetic elements (MGEs) in shaping the spread of this gene family and of AMR genes more broadly (32).

Molecular analyses point to the origins of $bla_{NDM}$ within an *Acinetobacter* background (4). Retrospective work on bacterial archives, including the present study, supports this claim, with growing evidence pointing to NbA as the likely source of $bla_{NDM}$. The fact that the gene was found in *A. junii* further heightens the probability that $bla_{NDM}$

originated in bacteria with a predominantly environmental footprint, a phenomenon that has been observed with numerous other antibiotic resistance genes (ARGs) of significant concern to human health, such as $bla_{CTX-M}$, $bla_{OXA-23}$, and $bla_{OXA-48}$ (33–36).

A series of events occurring over the last decades contributed to the pivotal role of *Acinetobacter* spp. in the $bla_{NDM}$ story: i) a shifting chemical milieu, both in the clinic and in the environment, followed by the introduction of carbapenem antibiotics in the 1980s; ii) the potential of many *Acinetobacter* spp. to exhibit resistance to carbapenems via intrinsic and acquired $bla_{OXA}$ genes, allowing them to gain a greater foothold in these chemically evolving settings, notably in hospitals; iii) the resulting increasing overlap of the *Acinetobacter* mobilome with more common human pathogens, namely, Enterobacterales.

Despite bottlenecks in the transmission and maintenance of MGEs from *Acinetobacter* spp. to Enterobacterales, these shifts likely contributed to potentiating the spread of $bla_{NDM}$ between these phylogenetically distant bacteria, if only by increasing their co-occurrences in highly selective environments. pNDM-BJ01-like plasmids, the only NDM-positive plasmids common to *Acinetobacter* spp. and Enterobacterales, contributed to this spread. Conjugation experiments with pNDM-BJ01-like plasmids nearly identical to pNDM-Ajun-H1-3 have been undertaken in three previous studies (4, 24, 37), demonstrating the transfer from *Acinetobacter* spp. to *E. coli*. When assessed, conjugation frequencies were more efficient between various *Acinetobacter* spp. as compared to those observed from *Acinetobacter* spp. to *E. coli* (37). Acquisition of $bla_{NDM-1}$ following conjugation experiments with pNDM-BJ01-like plasmids from *Acinetobacter* spp. to *E. coli* was confirmed by PCR in two studies (4, 24), with one group having sequenced the $bla_{NDM-1}$-containing plasmid in both the *A. baumannii* donor and the *E. coli* transconjugant (24). The latter carried $bla_{NDM-1}$ on a pNDM-BJ01-like plasmid with no insertions or additional sequences (24).

The *in vitro* experiments cited earlier (4, 24, 37) as well as our conjugation experiments results confirm the conjugative potential of pNDM-BJ01-like plasmids from *Acinetobacter* spp. to *E. coli*. Taken together with the two reported cases of pNDM-BJ01-like plasmids among clinical Enterobacterales isolates (23, 24), these experiments demonstrate a pathway of $bla_{NDM}$ spread from *Acinetobacter* spp. to Enterobacterales. In all these instances, $bla_{NDM}$ was located within a Tn*125* transposon, itself embedded on a pNDM-BJ01-like plasmid. Tn*125* has been proposed as the ancestral transposon responsible for the mobilization of $bla_{NDM}$ and is the only $bla_{NDM}$-containing transposon common to both *Acinetobacter* spp. and Enterobacterales (20). In the latter bacterial order, it has been primarily found among common plasmid replicon types (IncFII/FI; IncC; IncFIB/IncHI1B) and in fewer instances on the bacterial chromosome (20).

The genetic structure of the Tn*125*-like transposon we report, containing three full copies of IS*Aba125* and the *aph(3')-VIa* resistance gene, is quite rare (only two other cases have been reported in *Acinetobacter* species isolated 13 and 17 years later). The downstream part of this transposon made up of two copies of IS*Aba125* and the *aph(3')-VIa* resistance gene forms a complete Tn*aphA6* transposon (38). The presence of Tn*aphA6* at the tail-end of this Tn*125*-like transposon may provide insights into the genetic steps leading to the emergence of $bla_{NDM}$. Indeed, some have suggested that $bla_{NDM}$ is a chimera constructed following genetic recombination involving IS*Aba125* and the *aph(3')-VIa* resistance gene since the intergenic region between IS*Aba125* and $bla_{NDM}$ as well as the first 19 bp of $bla_{NDM}$ are identical to the intergenic region between IS*Aba125* and *aph(3')-VIa* as well as the first 19 bp of *aph(3')-VIa* (39). The Tn*125*-like transposon we report may in fact represent an intermediary genetic structure between the emergence of $bla_{NDM}$ and the Tn*125* transposon structure containing only two copies of IS*Aba125* (full or partial) that are more commonly found in *Acinetobacter* species, and particularly on BJ01-like plasmids.

A recent study estimated that $bla_{NDM}$ emerged prior to "1990, and possibly well back into the mid-twentieth century" on a Tn*125* transposon (20). The authors maintain that the Indian subcontinent remains a likely scenario of $bla_{NDM}$ origin in spite of their

modeling pointing to its emergence decades prior to its earliest known presence in India in 2005 and despite at least one unsuccessful attempt at recovering NDM-positive isolates among carbapenem-resistant strains isolated in India prior to 2005 (40). Our findings caution against premature conclusions as to $bla_{NDM}$'s origins (in India, Israel, or elsewhere), the understanding of which is still incomplete. The socio-environmental conditions leading to $bla_{NDM}$'s emergence may have been quite different from those that contributed to its dissemination, as has been demonstrated in other instances (41–43). The scarcity of data entails that $bla_{NDM}$ could very well have originated elsewhere than in India, before disseminating widely within its borders, in neighboring countries, and globally. Finally, our finding of this gene in a region distant from New Delhi, India (the city referenced in the gene's name), and prior to its earliest observations there, highlights the pitfalls of using geographic locations in selecting scientific names and reinforces the research community's stated objective of avoiding this practice (https://www.who.int/publications/i/item/WHO-HSE-FOS-15.1).

In conclusion, the identification of a $bla_{NDM-1}$-positive strain from Israel in 2004, predating what were previously the earliest known isolates from India in 2005, supports the need for further work in investigating the origins of this widespread and problematic ARG. Specifically, this study suggests that focusing on NbA and on environmental archives from the 20th and early 21$^{st}$ centuries in the Middle East and beyond may yield additional clues. Our observations reported here also strengthen the view that MGEs play significant roles in ARG emergence and spread (32). Understanding the origins and dissemination of ARGs sheds light on the processes promoting AMR, impacting policies aimed at mitigating its drivers and at preventing similar issues from arising in the future.

## ACKNOWLEDGMENTS

The authors would like to acknowledge and thank Chaim Sheba Medical Center, JMI Laboratories, and International Health Management Associates Inc. as the sources from which the Israeli isolates were obtained. We would also like to thank the Barcelona Institute for Global Health Foundation for sharing two NDM-positive strains isolated in Turkey (JVAP01 and JVAP02).

This work was supported by the New Frontiers in Research Fund, Canada (NFRFE-2019–00444) (L.-P.H. and H.L.L.), the Fonds de Recherche du Québec – Santé (282182) (L.-P.H.), and the Canadian Institute for Advanced Research (CIFAR) (GS-0000000256) (L.-P.H.).

## AUTHOR AFFILIATIONS

[1]Department of Biology, Faculty of Science, Université de Sherbrooke, Sherbrooke, Sherbrooke, Québec, Canada
[2]Department of Microbiology and Infectious Diseases, Faculty of Medicine and Health Sciences, Université de Sherbrooke, Sherbrooke, Canada
[3]CIUSSS de l'Estrie - CHUS, Sherbrooke, Québec, Canada
[4]Conflict and Health Research Group, King's College London, London, United Kingdom
[5]Institute for Society and Genetics, UCLA, Los Angeles, California, USA
[6]Institute for Medical Microbiology, Immunology and Hygiene, Faculty of Medicine and University Hospital Cologne, University of Cologne, Cologne, Germany
[7]German Centre for Infection Research (DZIF), partner site Cologne-Bonn, Cologne, Germany
[8]Centre de recherche Charles-Le Moyne, CISSS Montérégie-Centre, Longueuil, Québec, Canada

## AUTHOR ORCIDs

Simon Lévesque http://orcid.org/0000-0002-4356-9702
Paul G. Higgins http://orcid.org/0000-0001-8677-9454
Sébastien Rodrigue http://orcid.org/0000-0002-5366-7234

Louis-Patrick Haraoui  http://orcid.org/0000-0002-3713-7866

## FUNDING

| Funder | Grant(s) | Author(s) |
| --- | --- | --- |
| New Frontiers in Research Fund | NFRFE-2019-00444 | Louis-Patrick Haraoui |
| New Frontiers in Research Fund | NFRFE-2019-00444 | Hannah L. Landecker |
| FRQ \| Fonds de Recherche du Québec - Santé (FRQS) | 282182 | Louis-Patrick Haraoui |
| Canadian Institute for Advanced Research (ICRA) | GS-0000000256 | Louis-Patrick Haraoui |

## AUTHOR CONTRIBUTIONS

Frédéric Grenier, Data curation, Formal analysis, Methodology, Project administration, Software, Validation, Visualization, Writing – original draft | Vincent Baby, Data curation, Formal analysis, Methodology, Project administration, Software, Validation, Visualization, Writing – original draft | Sarah Allard, Data curation, Formal analysis, Writing – review and editing | Simon Lévesque, Formal analysis, Supervision, Validation, Writing – review and editing | François Papale, Formal analysis, Validation, Writing – review and editing | Richard Sullivan, Methodology, Writing – review and editing | Hannah L. Landecker, Funding acquisition, Writing – review and editing | Paul G. Higgins, Formal analysis, Validation, Writing – review and editing | Sébastien Rodrigue, Formal analysis, Methodology, Supervision, Validation, Writing – review and editing | Louis-Patrick Haraoui, Conceptualization, Data curation, Formal analysis, Funding acquisition, Investigation, Methodology, Project administration, Resources, Supervision, Validation, Writing – original draft, Writing – review and editing

## DATA AVAILABILITY

Sequencing reads are deposited in GenBank under BioProject number PRJNA1079318.

## ADDITIONAL FILES

The following material is available online.

Open Peer Review

**PEER REVIEW HISTORY (review-history.pdf).** An accounting of the reviewer comments and feedback.

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
