## [Reviewer comments · Microbiology Spectrum]

Microbiology Spectrum

Isolation of a *bla*_{NDM-1}-positive strain in Israel predating the earliest observations from India

Frédéric Grenier, Vincent Baby, Sarah Allard, Simon Lévesque, François Papale, Richard Sullivan, Hannah Landecker, Paul Higgins, Sébastien Rodrigue, and Louis-Patrick Haraoui

Corresponding Author(s): Louis-Patrick Haraoui, Université de Sherbrooke

Review Timeline:

Submission Date:	April 25, 2024
Editorial Decision:	June 10, 2024
Revision Received:	July 30, 2024
Accepted:	August 28, 2024

Editor: Ana-Maria Dragoi

Reviewer(s): Disclosure of reviewer identity is with reference to reviewer comments included in decision letter(s). The following individuals involved in review of your submission have agreed to reveal their identity: Nana Ama Amissah (Reviewer #1)

Transaction Report:

DOI: <https://doi.org/10.1128/spectrum.01002-24>

Re: Spectrum01002-24 (Isolation of a *bla*_{NDM-1}-positive strain in Israel predating the earliest observations from India)

Dear Dr. Louis-Patrick Haraoui:

Thank you for the privilege of reviewing your work. Below you will find my comments, instructions from the Spectrum editorial office, and the reviewer comments.

The reviewers finalized their recommendations and while largely positive, they raised a few questions that will need to be addressed before we accept the manuscript for publication. Please address the reviewers comments (Reviewer 1 had added the corrections directly on the manuscripts, please see attachment).

Revision Guidelines

Sincerely,
Ana-Maria Dragoi
Editor
Microbiology Spectrum

Reviewer #1 (Public repository details (Required)):

Sequenced data

Reviewer #2 (Comments for the Author):

This study showed an interesting result that never seen before. As blaNDM spreading almost all over the world, the research result in this study will give a necessary description for the origin of blaNDM. Overall, it's a detailed study. There are only a few questions that I am curious about:

1. Although the earliest known strain carrying blaNDM was obtained from Israeli patients, is it somewhat insufficient to search for only Israeli strains when searching for the earliest strains of possible origin of blaNDM? That is, there is not a single strong reason to justify the only possibility that only Israel is the earliest origin of the NDM? Is it necessary to align the blaNDM of known strains in the NCBI database?

2. In this paper, some of the plasmid conjugation and transfer experiments in the known research results are listed, so is it necessary to verify the conjugation ability of the plasmid pNDM-Ajun-H1-3 in this study?

**Full title** **Isolation of a *bla*_{NDM-1}-positive strain in Israel**
**predating the earliest observations from India**

**Short title** **Earliest observation of a *bla*_{NDM-1}-positive strain**

Frédéric Grenier¹, Vincent Baby¹, Sarah Allard², Simon Lévesque^{2,3}, François Papale²,
Richard Sullivan⁴, Hannah L. Landecker⁵, Paul G. Higgins^{6,7}, Sébastien Rodrigue¹ &
Louis-Patrick Haraoui^{*2,8}

1 Department of Biology, Faculty of Science, Université de Sherbrooke, Sherbrooke,
Québec, Canada

2 Department of Microbiology and Infectious Diseases, Faculty of Medicine and Health
Sciences, Université de Sherbrooke

3 CIUSSS de l'Estrie - CHUS, Sherbrooke, Québec, Canada

4 Conflict and Health Research Group, King's College London, London, United Kingdom

5 Institute for Society and Genetics, UCLA, Los Angeles, California, United States

6 Institute for Medical Microbiology, Immunology and Hygiene, Faculty of Medicine and
University Hospital Cologne, University of Cologne, Cologne, Germany

7 German Centre for Infection Research (DZIF), partner site Cologne-Bonn, Cologne,
Germany

8 Centre de recherche Charles-Le Moyne, CISSS Montérégie-Centre, Longueuil,
Québec, Canada

* **Corresponding author:** Louis-Patrick Haraoui

**Email:** louis.patrick.haraoui@usherbrooke.ca (LPH)

**Author Contributions:**

FG: Data curation, Formal Analysis, Methodology, Project Administration, Software,
Validation, Visualization, Writing - original draft.

VB: Data curation, Formal Analysis, Methodology, Project Administration, Software,
Writing - review & editing.

SA: Data curation, Writing - review & editing.

SL: Formal Analysis, Supervision, Validation, Writing - review and editing.

FP: Formal Analysis, Writing - review and editing.

RS: Methodology, Writing - review & editing.

HLL: Funding Acquisition, Writing - review & editing.

PGH: Formal Analysis, Validation, Writing - review & editing.

[revised manuscript text omitted]

copies of IS*Aba125*, as has often been reported among the earliest NDM-positive
isolates [17]. A third copy of IS*Aba125* was found on the same 49kb plasmid (exact size:
49,163 base pairs). This plasmid was found to match numerous other NDM-positive
plasmids termed pNDM-BJ01-like, based on their similarity to a plasmid initially reported
in an *A. Iwoffii* strain from November 2010 in China [18]. Since this initial description,
pNDM-BJ01-like plasmids have been shown to be widely circulating among
*Acinetobacter* spp., particularly among non-*baumannii* *Acinetobacter* (NbA), including:
pNDM-40-1 in an *A. bereziniae* from India from 2005 [4]; pNDM-JVAP02 and pNDM-

JVAP01, included in this report (from an *A. pittii* and an *A. lactucae* isolated in Turkey in
2006 and 2009 respectively) [8,9]; numerous *Acinetobacter* spp. from China from 2009
onwards [19], as well as elsewhere around the world. pNDM-BJ01-like plasmids have
also been identified in Enterobacterales clinical isolates in China and in Mexico in 1
*Klebsiella aerogenes* and 1 *Providencia rettgeri* respectively [20,21].

A BLAST search using the nr/nt database combined with a literature review (April 2,
2024) revealed a total of 51 BJ01-like plasmids [20]. Of these, 46 carried bla_{NDM} among
36 NbA, 8 *A. baumannii*, 1 *Klebsiella aerogenes* and 1 *Providencia rettgeri*. These
included 42 bla_{NDM-1} among 32 NbA, 8 *A. baumannii*, 1 *Klebsiella aerogenes* and 1
*Providencia rettgeri*; 2 strains isolated in South Korea in 2015 (CP137748.1 and
OR508935.1) reported as carrying bla_{NDM-1} but whose bla_{NDM} sequence does not
correspond to any known bla_{NDM} variant, likely constituting a new bla_{NDM} variant as they
each contain an identical 1 amino acid difference to bla_{NDM-1} (Arg81Lys; 1 *Acinetobacter*

[revised manuscript text omitted]

being present in pNDM-Ajun-H1-3 [17]. The authors maintain the Indian subcontinent
remains a likely scenario of *bla*_{NDM} origin in spite of their modelling pointing to its
emergence decades prior to its earliest known presence in India in 2005 and despite at
least one unsuccessful attempt at recovering NDM-positive isolates among carbapenem-
resistant strains isolated in India prior to 2005 [35]. Our findings caution against
premature conclusions as to *bla*_{NDM}'s origins, understanding of which is still incomplete.
The socio-environmental conditions leading to *bla*_{NDM}'s emergence may have been quite
different from those that contributed to its dissemination, as has been demonstrated in
other instances [36–38]. The scarcity of data entails that *bla*_{NDM} could very well have
originated elsewhere than in India, before disseminating widely within its borders, in
neighbouring countries and globally. Finally, our finding of this gene in a region distant
from New Delhi, India (the city referenced in the gene's name), and prior to its earliest
observations there positively contributes to the research community's efforts in selecting
scientific names (<https://www.who.int/publications/i/item/WHO-HSE-FOS-15.1>).

In conclusion, the identification of a *bla*_{NDM-1}-positive strain from Israel in 2004, predating
what were previously the earliest known isolates from India in 2005, supports the need
for further work in investigating the origins of this widespread and problematic ARG.
Specifically, this study suggests that focusing on NbA and on environmental archives
from the 20th and early 21st centuries may yield additional clues. Our observations
reported here also strengthen the view that MGEs play significant roles in ARG
emergence and spread [29]. Understanding the origins and dissemination of ARGs

sheds light on the processes promoting AMR, impacting policy aimed at mitigating its
drivers and at preventing similar issues from arising in the future.

**Acknowledgments**

The authors would like to acknowledge and thank Chaim Sheba Medical Center, JMI
Laboratories, and International Health Management Associates inc. as the sources from
which the Israeli isolates were obtained. We would also like to thank the Barcelona
Institute for Global Health Foundation for sharing two NDM-positive strains isolated in
Turkey (JVAP01 and JVAP02).

This work was supported by the New Frontiers in Research Fund, Canada (NFRFE-
2019-00444) (L.-P.H. and H.L.L.), the Fonds de Recherche du Quebec – Santé
(282182) (L.-P.H.) and the Canadian Institute for Advanced Research (CIFAR) (GS-
0000000256) (L.-P.H.).

**References**

- 1. Murray CJ, Ikuta KS, Sharara F, Swetschinski L, Robles Aguilar G, Gray A, et al.
Global burden of bacterial antimicrobial resistance in 2019: a systematic analysis. *The*
*Lancet*. 2022;399: 629–655. doi:10.1016/S0140-6736(21)02724-0
- 2. Tacconelli E, Carrara E, Savoldi A, Harbarth S, Mendelson M, Monnet DL, et al.
Discovery, research, and development of new antibiotics: the WHO priority list of
antibiotic-resistant bacteria and tuberculosis. *The Lancet Infectious Diseases*. 2018;18:
318–327. doi:10.1016/S1473-3099(17)30753-3
- 3. Logan LK, Weinstein RA. The Epidemiology of Carbapenem-Resistant
Enterobacteriaceae: The Impact and Evolution of a Global Menace. *The Journal of*
*Infectious Diseases*. 2017;215: S28–S36. doi:10.1093/infdis/jiw282
- 4. Jones LS, Carvalho MJ, Toleman MA, White PL, Connor TR, Mushtaq A, et al.
Characterization of Plasmids in Extensively Drug-Resistant *Acinetobacter* Strains
Isolated in India and Pakistan. *Antimicrob Agents Chemother*. 2015;59: 923–929.
doi:10.1128/AAC.03242-14
- 5. Jones LS, Toleman MA, Weeks JL, Howe RA, Walsh TR, Kumarasamy KK.
Plasmid Carriage of *bla*_{NDM-1} in Clinical *Acinetobacter baumannii* Isolates from India.
*Antimicrob Agents Chemother*. 2014;58: 4211–4213. doi:10.1128/AAC.02500-14
- 6. Hamidian M, Nigro SJ. Emergence, molecular mechanisms and global spread of
carbapenem-resistant *Acinetobacter baumannii*. *Microbial Genomics*. 2019;5.
doi:10.1099/mgen.0.000306
- 7. Simhon A, Rahav G, Shazberg G, Block C, Bercovier H, Shapiro M.
*Acinetobacter baumannii* at a Tertiary-Care Teaching Hospital in Jerusalem, Israel. *J*
*Clin Microbiol*. 2001;39: 389–391. doi:10.1128/JCM.39.1.389-391.2001

- 8. Roca I, Mosqueda N, Altun B, Espinal P, Akova M, Vila J. Molecular
characterization of NDM-1-producing *Acinetobacter pittii* isolated from Turkey in 2006.
*Journal of Antimicrobial Chemotherapy*. 2014;69: 3437–3438. doi:10.1093/jac/dku306
- 9. Cosgaya C, Mari-Almirall M, Van Assche A, Fernández-Orth D, Mosqueda N,
Telli M, et al. *Acinetobacter dijkshoorniae* sp. nov., a member of the *Acinetobacter*
*calcoaceticus*–*Acinetobacter baumannii* complex mainly recovered from clinical samples
in different countries. *International Journal of Systematic and Evolutionary Microbiology*.
2016;66: 4105–4111. doi:10.1099/ijsem.0.001318
- 10. Chen S, Zhou Y, Chen Y, Gu J. fastp: an ultra-fast all-in-one FASTQ
preprocessor. *Bioinformatics*. 2018;34: i884–i890. doi:10.1093/bioinformatics/bty560
- 11. Wick RR, Judd LM, Gorrie CL, Holt KE. Unicycler: Resolving bacterial genome
assemblies from short and long sequencing reads. Phillippy AM, editor. *PLoS Comput*
*Biol*. 2017;13: e1005595. doi:10.1371/journal.pcbi.1005595
- 12. Wood DE, Lu J, Langmead B. Improved metagenomic analysis with Kraken 2.
*Genome Biol*. 2019;20: 257. doi:10.1186/s13059-019-1891-0
- 13. Bortolaia V, Kaas RS, Ruppe E, Roberts MC, Schwarz S, Cattoir V, et al.
ResFinder 4.0 for predictions of phenotypes from genotypes. *Journal of Antimicrobial*
*Chemotherapy*. 2020;75: 3491–3500. doi:10.1093/jac/dkaa345
- 14. Naas T, Oueslati S, Bonnin RA, Dabos ML, Zavala A, Dortet L, et al. Beta-
lactamase database (BLDB) – structure and function. *Journal of Enzyme Inhibition and*
*Medicinal Chemistry*. 2017;32: 917–919. doi:10.1080/14756366.2017.1344235
- 15. Seemann T. Prokka: rapid prokaryotic genome annotation. *Bioinformatics*.
2014;30: 2068–2069. doi:10.1093/bioinformatics/btu153

- 16. Ankenbrand MJ, Hohlfeld S, Hackl T, Förster F. AliTV—interactive visualization
of whole genome comparisons. PeerJ Computer Science. 2017;3: e116.
doi:10.7717/peerj-cs.116
- 17. Acman M, Wang R, van Dorp L, Shaw LP, Wang Q, Luhmann N, et al. Role of
mobile genetic elements in the global dissemination of the carbapenem resistance gene
bla_{NDM}. Nat Commun. 2022;13: 1131. doi:10.1038/s41467-022-28819-2
- 18. Hu H, Hu Y, Pan Y, Liang H, Wang H, Wang X, et al. Novel Plasmid and Its
Variant Harboring both a *bla*_{NDM-1} Gene and Type IV Secretion System in Clinical
Isolates of *Acinetobacter lwoffii*. Antimicrob Agents Chemother. 2012;56: 1698–1702.
doi:10.1128/AAC.06199-11
- 19. Fu Y, Liu L, Li X, Chen Y, Jiang Y, Wang Y, et al. Spread of a common bla_{NDM}-
1-carrying plasmid among diverse *Acinetobacter* species. Infection, Genetics and
Evolution. 2015;32: 30–33. doi:10.1016/j.meegid.2015.02.020
- 20. Chen Z, Li H, Feng J, Li Y, Chen X, Guo X, et al. NDM-1 encoded by a pNDM-
BJ01-like plasmid p3SP-NDM in clinical *Enterobacter aerogenes*. Front Microbiol.
2015;6. doi:10.3389/fmicb.2015.00294
- 21. Marquez-Ortiz RA, Haggerty L, Olarte N, Duarte C, Garza-Ramos U, Silva-
Sanchez J, et al. Genomic Epidemiology of NDM-1-Encoding Plasmids in Latin
American Clinical Isolates Reveals Insights into the Evolution of Multidrug Resistance.
Genome Biology and Evolution. 2017;9: 1725–1741. doi:10.1093/gbe/evx115
- 22. Nemeč A, Španělová P, Shestivska V, Radolfová-Křížová L, Maixnerová M, Feng
Y, et al. Proposal for *Acinetobacter higginsii* sp. nov. to accommodate organisms of
human clinical origin previously classified as *Acinetobacter* genomic species 16.
International Journal of Systematic and Evolutionary Microbiology. 2023;73.
doi:10.1099/ijsem.0.006114

- 23. Zou D, Huang Y, Zhao X, Liu W, Dong D, Li H, et al. A Novel New Delhi Metallo-
β -Lactamase Variant, NDM-14, Isolated in a Chinese Hospital Possesses Increased
Enzymatic Activity against Carbapenems. *Antimicrob Agents Chemother.* 2015;59:
2450–2453. doi:10.1128/AAC.05168-14
- 24. Yong D, Toleman MA, Giske CG, Cho HS, Sundman K, Lee K, et al.
Characterization of a New Metallo- β -Lactamase Gene, *bla*_{NDM-1}, and a Novel
Erythromycin Esterase Gene Carried on a Unique Genetic Structure in *Klebsiella*
*pneumoniae* Sequence Type 14 from India. *Antimicrob Agents Chemother.* 2009;53:
5046–5054. doi:10.1128/AAC.00774-09
- 25. Wu W, Feng Y, Tang G, Qiao F, McNally A, Zong Z. NDM Metallo- β -Lactamases
and Their Bacterial Producers in Health Care Settings. *Clin Microbiol Rev.* 2019;32.
doi:10.1128/CMR.00115-18
- 26. Sun Y, Ji X, Liu Y, Liu Q, Guo X, Liu J, et al. New Delhi metallo- β -lactamase-1-
producing acinetobacter lwoffii of companion animal origin in China. *Indian Journal of*
*Medical Microbiology.* 2015;33: 615–617. doi:10.4103/0255-0857.167333
- 27. Wang Y, Wu C, Zhang Q, Qi J, Liu H, Wang Y, et al. Identification of New Delhi
Metallo- β -lactamase 1 in Acinetobacter lwoffii of Food Animal Origin. Cloeckaert A,
editor. *PLoS ONE.* 2012;7: e37152. doi:10.1371/journal.pone.0037152
- 28. McCann CM, Christgen B, Roberts JA, Su J-Q, Arnold KE, Gray ND, et al.
Understanding drivers of antibiotic resistance genes in High Arctic soil ecosystems.
*Environment International.* 2019;125: 497–504. doi:10.1016/j.envint.2019.01.034
- 29. David S, Cohen V, Reuter S, Sheppard AE, Giani T, Parkhill J, et al. Integrated
chromosomal and plasmid sequence analyses reveal diverse modes of carbapenemase
gene spread among *Klebsiella pneumoniae*. *Proc Natl Acad Sci USA.* 2020;117: 25043–
25054. doi:10.1073/pnas.2003407117

- 30. Humeniuk C, Arlet G, Gautier V, Grimont P, Labia R, Philippon A. β -Lactamases
of *Kluyvera ascorbata*, Probable Progenitors of Some Plasmid-Encoded CTX-M Types.
*ANTIMICROB AGENTS CHEMOTHER.* 2002;46: 5.
- 31. Poirel L, Figueiredo S, Cattoir V, Carattoli A, Nordmann P. *Acinetobacter*
*radioresistens* as a Silent Source of Carbapenem Resistance for *Acinetobacter* spp.
*Antimicrob Agents Chemother.* 2008;52: 1252–1256. doi:10.1128/AAC.01304-07
- 32. Poirel L, Heritier C, Nordmann P. Chromosome-Encoded Ambler Class D β -
Lactamase of *Shewanella oneidensis* as a Progenitor of Carbapenem-Hydrolyzing
Oxacillinase. 2004;48.
- 33. Surette MD, Wright GD. Lessons from the Environmental Antibiotic Resistome.
*Annu Rev Microbiol.* 2017;71: 309–329. doi:10.1146/annurev-micro-090816-093420
- 34. Huang T-W, Lauderdale T-L, Liao T-L, Hsu M-C, Chang F-Y, Chang S-C, et al.
Effective transfer of a 47 kb NDM-1-positive plasmid among *Acinetobacter* species. *J*
*Antimicrob Chemother.* 2015;70: 2734–2738. doi:10.1093/jac/dkv191
- 35. Castanheira M, Deshpande L, Woosley L, Prochaska R, Jones R. Retrospective
Search for NDM-1 Reveals Possible Indian Origin of DIM-1 Metallo-beta-lactamase.
*ECCMID 2012 - Poster 1240.* 2012.
- 36. Chin C-S, Sorenson J, Harris JB, Robins WP, Charles RC, Jean-Charles RR, et
al. The Origin of the Haitian Cholera Outbreak Strain. *N Engl J Med.* 2011;364: 33–42.
doi:10.1056/NEJMoa1012928
- 37. Buchholz U, Bernard H, Werber D, Böhmer MM, Remschmidt C, Wilking H, et al.
German Outbreak of *Escherichia coli* O104:H4 Associated with Sprouts. *N Engl J Med.*
2011;365: 1763–1770. doi:10.1056/NEJMoa1106482

38. Njamkepo E, Fawal N, Tran-Dien A, Hawkey J, Strockbine N, Jenkins C, et al.
Global phylogeography and evolutionary history of *Shigella dysenteriae* type 1. *Nat*
*Microbiol.* 2016;1: 16027. doi:10.1038/nmicrobiol.2016.27

A

**pNDM-Ajun-H1-3's
Tn125 transposon**

**B****No NDM**
July 28, 2024

Dear Editor,

We would like to thank both reviewers for having taken the time to read and comment our manuscript. Below, please find point-by-point responses to the issues raised by them.

We hope that with these modifications, the manuscript will be deemed satisfactory to be published in *Microbiology Spectrum*.

With best wishes,

Louis-Patrick Haraoui, MD, MSc, FRCPC (on behalf of all co-authors)
Associate Professor, Department of Microbiology and Infectious Diseases
Faculty of Medicine and Health Sciences, Université de Sherbrooke

Reviewer # 1 (Public repository details (Required)): Sequenced data

Response: Sequenced data are available in GenBank under BioProject number PRJNA1079318.

Comments embedded in the manuscript.

Comments 1 & 2: “Lines 84-87 contradicts statement made on lines 79-83.” “Please rephrase appropriately.”

Response: We do not perceive the contradiction pointed out by Reviewer #1 between the statement on lines 79-83 and the one on lines 84-87. In the former statement, we mention that NDM variants have become the most prevalent carbapenemase in the *Enterobacteriaceae* family of organisms. In the latter statement, we point out that only a minority of carbapenem-resistant *Acinetobacter baumannii* carry *bla*_{NDM}. Since *A. baumannii* is not classified in the *Enterobacteriaceae* family but rather in the *Moraxellaceae* family, we do not consider these two statements as contradictory. In other words, NDM variants are highly prevalent in one family of organisms (*Enterobacteriaceae*) and seldom found in a species, *A. baumannii*, which belongs to a distinct family of organisms (*Moraxellaceae*). If Reviewer #1 is pointing to another contradiction between those statements, we welcome their further input to address it.

Comment Line 91: “Please add more references to support your statement.”

Response: We have done so directly in the revised manuscript. Specifically, we have added the following references: Marchaim *et al.* (10.1007/s10096-008-0545-z) and Paul *et al.* (10.1016/j.jhin.2005.01.007).

Comment Line 152: “Please be consistent, which of the NDM variant?”

Response: We have updated the manuscript to specify we are referring to *bla*_{NDM-1}.

Comment Line 195: “please add a punctuation mark after the sentence.”

Response: The sentence runs from line 195 to line 197 and does not end at the end of line 195. To avoid any confusion, we have removed the capital letters from the words “accession number”.

Comment Lines 216-217: “The results section has not h”

Response: This comment is incomplete and we could not address it. If Reviewer #1 finds it essential to address something in this part of manuscript, we welcome their return.

Reviewer # 2 (Comments for the author)

Comment #1. Although the earliest known strain carrying *bla*_{NDM} was obtained from Israeli patients, is it somewhat insufficient to search for only Israeli strains when searching for the earliest strains of possible origin of *bla*_{NDM}? That is, there is not a single strong reason to justify the only possibility that only Israel is the earliest origin of the NDM? Is it necessary to align the *bla*_{NDM} of known strains in the NCBI database.

Response: We agree with Reviewer # 2 that it is insufficient to search for the possible origin of *bla*_{NDM} only in Israeli strains, and it is certainly not our claim that *bla*_{NDM} “originated” in Israel. Rather, our paper aims to highlight that strain Ajun-H1-3 we identified and describe “muddies the waters” of the geographic origin story of *bla*_{NDM}. We tried to make that clear in lines 265-266 of the original manuscript in which we state: “Our findings caution against premature conclusions as to *bla*_{NDM}’s origins, understanding of which is still incomplete.” To reinforce this point and Reviewer # 2’s comment, we have modified that sentence in the updated version of the manuscript. Moreover, we had already mentioned in the final paragraph of the manuscript that further sequencing of *Acinetobacter* spp. from the late 20th and early 21st century may yield additional clues to help piece together the origin story of *bla*_{NDM}. We also modified that sentence to make it explicit that this suggestion is not limited to isolates from Israel.

As for the last question of the first comment (aligning “the *bla*_{NDM} of known strains in the NCBI database”), we edited the manuscript and figure to reflect certain unique aspects of the transposon carrying *bla*_{NDM} in the strain we report.

2. In this paper, some of the plasmid conjugation and transfer experiments in the known research results are listed, so is it necessary to verify the conjugation ability of the plasmid pNDM-Ajun-H1-3 in this study?

Response: We performed conjugation experiments with pNDM-Ajun-H1-3 and present these results in the modified manuscript.

Re: Spectrum01002-24R1 (Isolation of a *bla*_{NDM-1}-positive strain in Israel predating the earliest observations from India)

Dear Dr. Louis-Patrick Haraoui:

Your manuscript has been accepted, and I am forwarding it to the ASM production staff for publication. Your paper will first be checked to make sure all elements meet the technical requirements. ASM staff will contact you if anything needs to be revised before copyediting and production can begin. Otherwise, you will be notified when your proofs are ready to be viewed.

Sincerely,
Ana-Maria Dragoi
Editor
Microbiology Spectrum